# In Vitro Maturation of Retinal Pigment Epithelium Is Essential for Maintaining High Expression of Key Functional Genes

**DOI:** 10.3390/ijms21176066

**Published:** 2020-08-23

**Authors:** Abdullah Al-Ani, Saud Sunba, Bilal Hafeez, Derek Toms, Mark Ungrin

**Affiliations:** 1Department of Comparative Biology and Experimental Medicine, Faculty of Veterinary Medicine, University of Calgary, Calgary, AB T2N 1N4, Canada; aalani@ucalgary.ca (A.A.-A.); saud.sunba@ucalgary.ca (S.S.); muhammad.hafeez@ucalgary.ca (B.H.); mdungrin@ucalgary.ca (M.U.); 2Alberta Children’s Hospital Research Institute, University of Calgary, Calgary, AB T2N 4N1, Canada; 3Alberta Diabetes Institute, University of Alberta, Edmonton, AB T6G 2E1, Canada; 4Biomedical Engineering Graduate Program, University of Calgary, Calgary, AB T2N 1N4, Canada; 5Leaders in Medicine Program, Cumming School of Medicine, University of Calgary, Calgary, AB T2N 4N1, Canada

**Keywords:** retinal pigment epithelium (RPE), maturation, differentiation, embryonic stem cells, pigment epithelium derived factor (PEDF), cell culture

## Abstract

Age-related macular degeneration (AMD) is the leading cause of blindness in the industrialized world. AMD is associated with dysfunction and atrophy of the retinal pigment epithelium (RPE), which provides critical support for photoreceptor survival and function. RPE transplantation is a promising avenue towards a potentially curative treatment for early stage AMD patients, with encouraging reports from animal trials supporting recent progression toward clinical treatments. Mature RPE cells have been reported to be superior, but a detailed investigation of the specific changes in the expression pattern of key RPE genes during maturation is lacking. To understand the effect of maturity on RPE, we investigated transcript levels of 19 key RPE genes using ARPE-19 cell line and human embryonic stem cell-derived RPE cultures. Mature RPE cultures upregulated PEDF, IGF-1, CNTF and BDNF—genes that code for trophic factors known to enhance the survival and function of photoreceptors. Moreover, the mRNA levels of these genes are maximized after 42 days of maturation in culture and lost upon dissociation to single cells. Our findings will help to inform future animal and human RPE transplantation efforts.

## 1. Introduction

Age-related macular degeneration (AMD) is the leading cause of blindness in the industrialized world, affecting more than 20% of adults over the age of 65 [1]. Visual impairment leads to a significantly reduced quality of life and accounts for billions of dollars in direct and indirect annual healthcare costs in the USA alone [2,3,4]. The earliest signs of AMD are often abnormalities of the retinal pigmented epithelium (RPE) monolayer and accumulation of drusen (complex, poorly understood protein/lipid deposits) immediately adjacent to it. In dry (or geographic) AMD, which accounts for ~90% of AMD patients, RPE dysfunction and atrophy is followed by a local, progressive and irreversible loss of photoreceptors [5,6]. In wet (or neovascular) AMD, acute penetration of the choroidal vasculature through the RPE into the eye results in rapid vision loss

The RPE secretes cytokines and growth factors essential for maturation and survival of both the retina and the choroid, and is also responsible for maintaining a barrier between these compartments, ensuring that choroidal signals (such as vascular endothelial growth factor, VEGF) do not enter the retina, and vice versa [7,8,9,10]. RPE cells also maintain photoreceptor survival and function directly, phagocytosing damaged outer segments, transporting nutrients and waste products and regenerating visual pigment [11]. Thus, it is reasonable to hypothesize that loss of normal RPE function is a central driver of AMD.

RPE transplantation has been shown to rescue photoreceptors from death in the RCS rat model of retinal degeneration that is caused by RPE atrophy [12,13,14,15,16]. This is an attractive approach to treat AMD as we are capable of efficiently differentiating stem cells into RPE for transplantation [17,18,19], and clinical trials are now underway [20,21,22,23,24,25]. With RPE cell therapy rapidly making its way to the clinic, important concerns regarding the exact phenotype of the transplanted RPE remain [26].

One such consideration is RPE maturation, which has been reported to enhance the function of RPE from various sources including: a primary cell-derived human cell line (ARPE-19) [27,28]; embryonic stem cells [20,29,30,31]; induced pluripotent stem cells [23] and adult stem cells [26]. ARPE-19 monolayers have been shown to exhibit substantially increased epithelial resistance and more physiological morphology after maturation for more than twenty days [27,28], while adult-stem cell-derived RPE cells demonstrated superior post-transplant function when transplanted into rats at the fourth week of differentiation [26]. RPE maturity has been reported to affect cellular morphology [27,28], pigmentation [23,32] and the production of several functional proteins such as pigment epithelium-derived factor (PEDF), bestrophin-1 (BEST1) and cellular retinaldehyde-binding protein (CRALBP) [20]. However, despite these demonstrations of the significance of RPE maturation, its molecular signature has not yet been established, nor is there a consensus on the culture duration required to achieve maturity in vitro.

In this study, we employed ARPE-19 and embryonic stem cell-derived (E-)RPE in vitro cultures to investigate the transcript profile of 19 genes (identified from the literature as important to RPE function) during maturation. We found increased transcript levels for 13 and 16 of these genes in matured ARPE-19 and E-RPE monolayers respectively, over nonmatured control cultures. ARPE-19 and E-RPE monolayers matured for 8 weeks demonstrated robust expression of these key functional genes which was largely and rapidly lost after single cell passaging. Our findings emphasize the potential benefits of transplanting RPE as a tissue rather than dissociated cells; allowing that tissue to mature prior to use and including confirmation of the molecular fingerprint of mature RPE tissue as part of clinical release criteria.

## 2. Results 

### 2.1. E-RPE Demonstrates Development in Morphology and Pigmentation as they Mature

Cellular morphology and pigmentation have been used in the field as qualitative criteria to evaluate RPE maturity and suitability for transplantation [20,23,26]. To assess the relationship of these phenotypes with maturation, ARPE-19 and E-RPE monolayers were cultured for up to 70 days and sampled at 14-day intervals. As E-RPE cultures mature, a distinct hexagonal or cobble-stone cell morphology progressively emerges over days 28–56, along with a parallel gain in pigmentation (Figure 1). ARPE-19 cultures were sampled on the same experimental timeline but, as previously reported, this spontaneously immortalized human cell line does not exhibit these markers of primary RPE [33].

### 2.2. E-RPE and ARPE-19 Cultures Demonstrate Progressive Increase in mRNA Levels of Key Functional RPE Genes as They Mature

As many transplantation studies use morphology and pigmentation as an indicator for RPE maturity, and ultimately utilize RPE after a range of culture durations [20,21,23], we investigated the mRNA levels of *PEDF*, *VEGF*, *PDGFA*, *CNTF*, *BDNF*, *FGF2*, *IGF1* and *RPE65* over time in culture. We focused our initial exploration on this subset of genes, as they have implications for various retinal degenerative diseases [34,35]. By day 42, most investigated E-RPE transcripts had attained 75% of their maximum level (5/8) with the greatest change taking place between four and 28 days of maturation (Figure 2). ARPE-19 cultures followed a similar trend, with six of eight investigated genes increasing with time. The exception is that unlike during E-RPE maturation, *VEGF* and *FGF2* mRNA levels in ARPE-19 progressively dropped with days in culture. 

### 2.3. Mature ARPE-19 Upregulates Transcript Levels of Key RPE Genes

After identifying 42 days as a critical point in RPE maturation, we expanded our analysis to 19 genes that code for proteins that carry essential and specific RPE functions such as photoreceptor support, choroid support, visual recycling and immune modulation for analysis (all investigated genes are described Table A1). Mature ARPE-19 significantly upregulates the mRNA levels of 13 of 19 investigated RPE functional genes (Figure 3a), including *PEDF*, *BDNF*, *CNTF*, *GAS6* and *IGF1*—genes that code for RPE-secreted factors known to enhance photoreceptor survival and function in retinal degenerative animal models [36,37,38,39]. Mature ARPE-19 also showed increased mRNA levels of genes that code basally secreted RPE factors known to contribute to the stability of the choroid, namely *PDGFA*, *FASL*, and *TGFB* [35,40,41,42]. Levels of some factors, including *VEGFA* and *FGF2*, were lower in mature ARPE-19. To confirm that the differential expression seen in mRNA levels was reflected by secreted proteins, we investigated the conditioned media taken from immature and mature cultures for levels of PEDF, PDGF-AA (a homodimer of two ‘A’ subunits coded for by *PDGFA*), VEGF-A and FGF-2. We observed that mature ARPE-19 secreted significantly more PEDF and PDGF-AA protein. Levels of secreted VEGF-A and FGF-2 appeared to decrease with maturation, but the change was not statistically significant (Figure 3b). 

### 2.4. Mature E-RPE Culture Upregulates Transcript Levels of Key RPE Genes

Compared to nonmatured control cultures, mature E-RPE cultures upregulate mRNA levels of 16 out of 19 investigated RPE genes, and significantly downregulate mRNA levels of *CXCL8* and *CCL2* (Figure 4a). As with ARPE-19, we investigated the conditioned media of both cultures and found mature RPE secreted significantly more PEDF, PDGF-AA and VEGF-A, and less FGF-2 when compared to immature culture (Figure 4b). Notably, the lower secretion of FGF-2 in mature ES-RPE cultures is not in line with the mRNA levels; it is in line with the trend in both mRNA and protein seen in mature ARPE-19 cultures (Figure 3). Finally, we compared the rank correlation of fold-change in transcript levels for mature cultures between ARPE-19 and E-RPE and found a strong relationship, suggesting that genes are similarly upregulated in both cell types (Spearman’s rank correlation coefficient, r_s_ = 0.7; *p* < 0.001).

### 2.5. Dissociation of the RPE Monolayer to Single Cell Suspension Resets Maturation

We next investigated the effect of dissociation on RPE cells, as this is a common method of transplantation [24,25]. To do this, we matured E-RPE for 56 days before dissociation and passaging, followed by another 14 days of maturation. Interestingly, after passaging, a portion of E-RPE cells quickly regained the hexagonal morphology observed on day 56 but nearly all cells quickly lost pigmentation (Figure 5a). Even at two weeks postpassage, many (5/8, 62.5%) transcripts showed levels comparable with those from RPE four days after initial seeding (Figure 5b). We observed that mRNA levels of *VEGF*, *IGF1* and *FGF2*, which did not significantly increase during maturation, changed little after passaging.

## 3. Discussion

While RPE maturation has been reported to enhance RPE function both in vitro and in vivo, it is not applied consistently and preparation processes for the RPE transplanted in animal and human trials vary widely [20,26,27,28,32]. To inform the selection and development of these processes, we have, therefore, characterized the impact of in vitro maturation on transcript levels and key secreted proteins in ARPE-19 and E-RPE.

We prospectively selected 19 genes playing important roles in photoreceptor stability, choroid stability, visual recycling, immune modulation and RPE-specific functions. This latter group included retinoid isomerhydrolase (RPE65), a crucial enzyme in the sequence of reactions that recycle 11-cis retinal in RPE cells [43]; and BEST1, which codes for a membrane protein uniquely expressed in RPE that functions as an anion channel and a regulator of intracellular calcium signaling [44]. All genes were selected prior to starting experiments, and none were subsequently excluded from, or added to, the analyses. While we were not able to perform a transcriptome-wide assay, such as RNA-seq to identify additional genes that may be changing in RPE during culture maturation, this has been successfully done by others to investigate RPE subjected to oxidative stress [45] and gene expression changes associated with pigmentation [32,46]. Such an approach would reinforce the work presented here and may reveal additional noncanonical genes that further contribute to the improved performance of mature RPE and should be considered for future experiments.

Human pluripotent stem cells are a reliable, sustainable and therapeutically relevant source for producing RPE that demonstrate enhanced gene expression, morphology and function [47,48] For this reason, they are currently being utilized in clinical trials to treat retinal degenerative diseases [20,21,49]. 

Upon maturation, both ARPE-19 and E-RPE upregulate mRNA levels of RPE genes known to be critical for their in vivo function (Figure 3 and Figure 4). Genetic variation in 14 of the 19 genes we investigated is associated with some form of retinal degenerative disease. In particular, the upregulation of *PEDF*, *CNTF*, *IGF1* and *BDNF* is of therapeutic value, as they are well-recognized neurotrophic factors secreted by RPE to support photoreceptors [7,37,50]. These factors have been shown to rescue photoreceptor survival and function in various retinal degeneration animal models [37,38,51], and have been identified as therapeutic targets for AMD and other retinal degenerative diseases [37,52,53,54]. We speculate that the upregulation we show here may at least partially explain the mechanism underlying the reported superiority of mature RPE in supporting photoreceptor survival [26,27,28]. 

Our observation of increased PEDF expression is also relevant for its effects on the choroid, where it has been demonstrated to prevent angiogenic retinopathies such as wet AMD by antagonizing VEGF signaling [55,56,57]. Increased VEGF expression by E-RPE over time (Figure 4) and particularly the high transcript levels following single-cell dissociation (Figure 5), suggest a spontaneous epithelial to mesenchymal transition, which has been observed in E-RPE cells that are routinely passaged [58]. Alternatively, it may relate to the fact that in vivo, RPE forms a well-defined boundary between the choroidal and retinal compartments—future studies focused on identifying the directionality of VEGF secretion may yield greater understanding. Mature ARPE-19 and E-RPE also both demonstrated increased transcription of choroid stability factors including *TGFB*, *TIMP3* and *PDGFA*. TGF-β is known to have an immune suppressive effect, which enables RPE to establish and maintain the immune privilege of the eye [59]. TIMP3 is secreted by RPE basally to remodel Bruch’s membrane, inhibit angiogenesis and regulate inflammation [60], and mutations and deficiencies in TIMP3 have been linked to various retinal degenerative diseases including AMD and Sorsby’s fundus dystrophy [61,62]. PDGF-AA is secreted by RPE basally to support the choroid and regulate angiogenesis [35]. In addition to supporting photoreceptors with trophic factors, we speculate that mature RPE may better regulate blood vessel growth and control neovascularization.

We also observed downregulation of *CCL2* and *CXCL8*, which code for monocyte chemoattractant protein (MCP)-1 and interleukin (IL)-8, respectively. Both are proinflammatory chemokines secreted by the RPE that act on immune cells and modulate the immune response under physiological and pathological conditions [63]. The expression of these proteins in healthy animals and individuals is quite low, and clinical studies have reported a direct correlation between their expression and incidence of AMD [64,65]. We speculate that these factors may recruit immune cells as a secondary line of defense in the event that the RPE is unable to provide a complete physical barrier between the choroid and retina, sacrificing potential inflammatory damage in favor of immediate response to injury or pathogens. This would explain their elevated expression in both immature and deteriorating RPE as compared to mature RPE. Further investigations are required to confirm specific roles played by these upregulated transcripts in the functional improvement observed in matured RPE cells.

Changes in gene expression during maturation were broadly similar between ARPE-19 and E-RPE, supporting a generalizable effect of RPE maturation on gene expression. However, *FGF2* and *VEGFA* mRNA levels did not follow this trend as they increased in mature E-RPE but decreased or remained stable in mature ARPE-19 cells (Figure 3 and Figure 4). This is not unexpected, as previous studies have identified differences in gene expression and behavior between ARPE-19, a spontaneously immortalized cell line, and E-RPE [18,27,33]. Moreover, we have observed that, unlike E-RPE, ARPE-19 cells do not exhibit the traditional hexagonal RPE morphology and pigmentation in culture, even after 70 days of maturation (data not shown). While our observation is consistent with the majority of literature accounts, newly emerging evidence from Samuel and colleagues indicate that ARPE-19 can adopt the traditional hexagonal RPE morphology when they are aged longer than 120 days in culture [46]. Our observations that ARPE-19 and E-RPE respond largely similarly to maturation do generally support the continued use of the ARPE-19 cell line model. One limitation of this study was the absence of endogenous RPE derived from adult donors, although others have shown considerable similarities in the transcriptome between both ARPE-19 and E-RPE, particularly after maturation [32,46]. 

The expression kinetics of *IGF1*, *PEDF* and *RPE65* in E-RPE are intriguing as they demonstrate increased expression as the culture ages up until 56 days (Figure 2). While it is possible that the expression of these genes has reached a maximum, additional time points beyond 70 days would be required to confirm this observation. Our findings are generally consistent with Da Cruz and colleagues who reported that PEDF secretion increases as RPE matures and then declines after ~80 days in culture [20]. The drop in PEDF, IGF-1 and RPE65 transcription after 56 days in E-RPE is, however, not seen in ARPE-19. While the expression of most examined genes in E-RPE peaked at 42–56 days, further in vivo investigations are required to determine if that would be the optimal maturation age for RPE transplantation. 

Given the reversion of maturation we observed following dissociation, we anticipate that the transplantation of RPE as multicellular tissues [20,21,23,31] will improve the effectiveness of treatments over single cell suspensions by maintaining RPE maturity. Indeed, transplantation of E-RPE monolayers (or “sheets”) survived better, and led to significant improvements in visual acuity and photoreceptor survival, compared to transplantation of single cell suspensions [29,31]. Future work would also benefit from an investigation of RPE tissue characteristics, like the establishment of tight junctions and subsequent increased transepithelial resistance, to determine whether the improved gene expression profile of mature RPE we observed requires assembly into a coherent tissue.

In this work, we demonstrated that the in vitro maturation of ARPE-19 and E-RPE leads to increased expression of genes crucial for a range of critical RPE functions. Matured RPE cultures had higher transcript levels of therapeutically relevant neurotrophic and choroid stability factors; increased production of PDGF-AA and PEDF proteins and lower transcript levels of cytokines that modulate the recruitment of immune cells to the retina. Taken together, our results provide plausible mechanisms for the reported superiority of mature RPE in therapeutic transplantation.

While further investigations are needed to conclusively relate maturation to clinical outcomes, based on our findings we recommend that RPE cultures be matured for at least 42 days for best in vitro and in vivo performance and that expression levels of factors such as PEDF, TIMP3 and CXCL8 be considered when developing release criteria for RPE batches intended for clinical use.

## 4. Materials and Methods

### 4.1. Tissue Culture

ARPE-19 cells were seeded in a 6-well plate (VWR, Mississauga, ON, Canada. cat # 82050-842) at 600,000 cells/well (60,000 cells/cm^2^) and reached confluency at ~120,000 cells/cm^2^. Cultures were maintained for up to 70 days in 2 mL of ARPE-19 culture media that consisted of: DMEM/F-12, HEPES (Thermo Fisher Scientific, Mississauga, ON, Canada. cat # 11330057), 10% FBS (VWR. cat # 97068-085), and 1% Pen/strip (Thermo Fisher cat # 15140122). Media was changed every 48 h by replacing the entire old media volume with 2mL of fresh media. For all experiments, E-RPE cells were cultured in an identical manner to ARPE-19 cells, using ES-RPE culture media that consisted of: 70% DMEM; 30% F12; 2% B-27 supplement and 1% Pen/Strip (Thermo Fisher. cat # 11965, 11765, 17504, and 15140122).

To assess the oxygenation status of the cells under the outlined culture conditions, oxygen delivery was calculated using our previously published method [66]. A RPE oxygen consumption rate of 42 amol∙cell^−1^∙s^−1^ was used for the calculation [67]. We determined that cultured RPE cells should be receiving an adequate amount of oxygen, with a local oxygen concentration of 1.42 × 10^−4^ mol/L at the cells and a maximum oxygen delivery rate of 191 amol∙cell^−1^∙s^−1^. 

### 4.2. RNA Extraction and cDNA Synthesis

Adherent cultures of ARPE-19 and E-RPE were collected by adding 1 mL of TrypLE Express Enzyme (Thermo. cat # 12604013) into each well and incubating for 5 min at room temperature. Next, collected samples were centrifuged for 5 min at 200× *g* and cell pellets were stored at −80 °C for up to one month before RNA isolation using a Total RNA Purification kit (Norgen Biotek Corp., Thorold, ON, Canada. cat # 37500). For each cDNA reaction, 1 µg RNA was used as input for the iScript Reverse Transcription Supermix kit (Bio-Rad Laboratories Ltd., Mississauga, ON, Canada. cat # 1708841). The cDNA was then used for real-time polymerase chain reaction (RT-qPCR) using PowerUp™ SYBR™ Green Master Mix (Thermo Fisher cat # A25777) to analyze and compare expression levels of selected genes in immature and mature RPE cultures using primer sequences detailed in Table A1. SYBR Green RT-qPCR was carried out with technical duplicates using the 7500 Fast Real-Time PCR System (Thermo Fisher) with *PPIA* as a housekeeper gene, as has been used previously [68,69]. Analyses were performed on either ∆CT values or ∆∆CT values where the reference sample was set at 4 days of culture.

### 4.3. Conditioned Media Analysis of Secreted Protein

Conditioned media (CM) was collected during the regular media change (every 48 h) for both ARPE-19 and E-RPE cells. The conditioned media was stored at −80 °C before being assayed. Multiplexing Laser Bead Technology (Eve Technologies, Calgary, AB, Canada) was performed on CM to estimate the concentration of proteins of interest. PEDF ELISA (Abcam, Cambridge, MA, USA. cat # ab213815) was utilized to measure the amount of PEDF in conditioned media from E-RPE as the original PEDF multiplex assay had been discontinued.

### 4.4. E-RPE Differentiation

RPE cells were differentiated from human embryonic stem cells as previously described by Maruotti and colleagues [70] with the following two modifications: (1) the HES-2 cell line was grown to confluence under 5% CO_2_ and 5% O_2_ in mTeSR1 and (2) during induced differentiation, a concentration of 50 nM chetomin was used. Cells were grown to a low passage (3–5) before being expanded and cryogenically stored.

### 4.5. Statistical Analysis

All experimental units (*n*) are biologically distinct repeats conducted from fresh thawed vials of banked cell lines and were repeated a minimum of three times. Data were analyzed with GraphPad Prism (v.7, San Diego, CA, USA) and R statistical software (v.3.5.1, Vienna, Austria [71]) Specific statistical methods are described in the text and figure legends. We considered *p*-values of less than 0.05 statistically significant. 

## Figures and Tables

**Figure 1 ijms-21-06066-f001:**
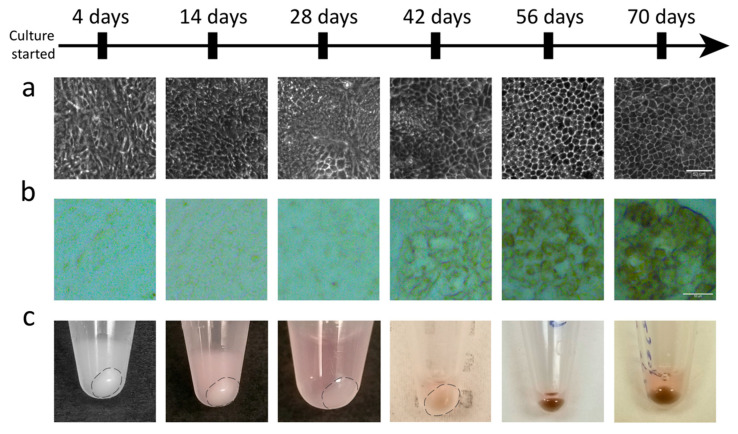
Embryonic stem cell-derived retinal pigment epithelium (E-RPE) cultures demonstrate morphological changes and become pigmented as they mature. (**a**) Cellular morphology of E-RPE cultures develops over time to make the consistent hexagonal shaped cells seen at 56 and 70 days. (**b**) Representative micrographs of the progressive pigmentation of the E-RPE culture as it matures. (**c**) Cell pellets in microcentrifuge tubes of E-RPE cultures sampled at confluency (4 days), 14 days, 28 days, 42 days, 56 days and 70 days show increasing pigmentation over time. Scale bars represent lengths of 50 µm in (**a**) and (**b**).

**Figure 2 ijms-21-06066-f002:**
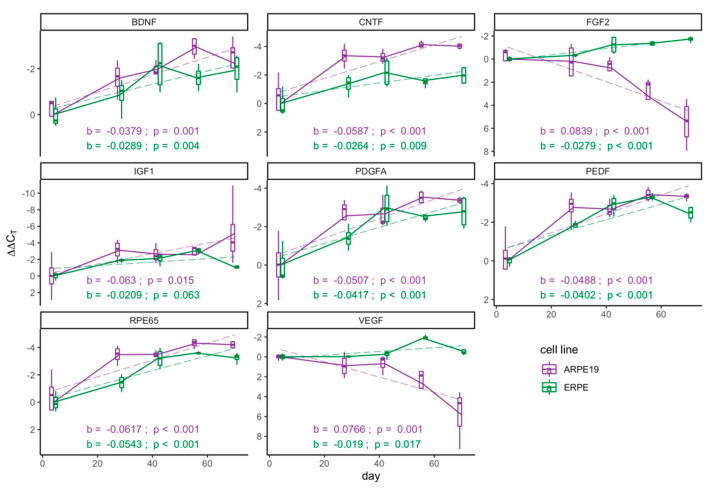
Key functional genes progressively increase with maturation. Real-time polymerase chain reaction (RT-qPCR) analysis of mRNA levels across 70 days of culture in E-RPE culture (green) and ARPE-19 culture (purple). Results were normalized to an endogenous reference gene (*PPIA*) and are presented as ΔΔCt means (*n* = 4) ± standard deviation at each time point. A linear regression model (dashed line) was used to describe the relationship between ΔΔCt values and days of maturation of each gene for both RPE cell sources; regression coefficients, denoted as b, and associated *p*-values, to reject the null hypothesis that b = 0, are shown on each graph.

**Figure 3 ijms-21-06066-f003:**
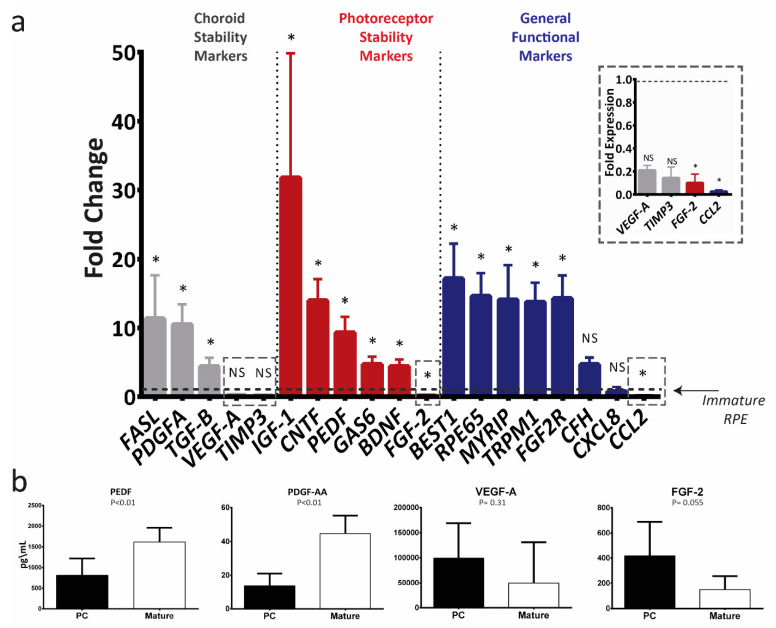
In vitro maturation of ARPE-19 upregulates the expression of therapeutically relevant photoreceptor and choroid trophic factors. (**a**) RT-qPCR analysis of mRNA levels of key RPE genes in mature ARPE-19 cells (>42 days in culture; *n* = 4). The results were normalized to an endogenous reference gene (*PPIA*) and are presented as mean fold change (2^−^^ΔΔCT^) relative to immature ARPE-19 cultures (dotted line) ± standard deviation. Data were compared with a Mann–Whitney U test of ΔCt values; * *p* < 0.05; NS: not significant. (**b**) Protein secreted by the post-confluent (PC) immature and mature ARPE-19 cultures into the conditioned media presented as mean concentration ± standard deviation (*n* = 5), Mann–Whitney U test.

**Figure 4 ijms-21-06066-f004:**
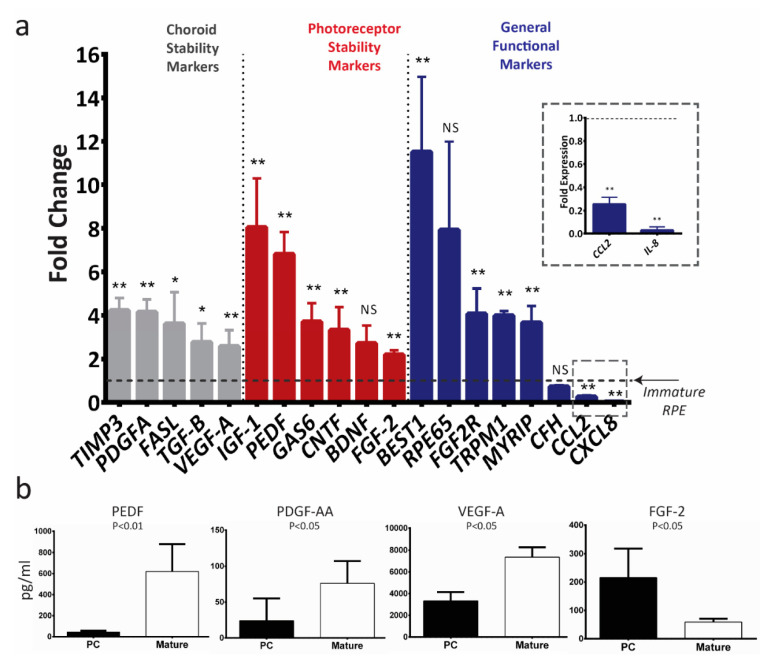
In vitro maturation of E-RPE cells upregulates the expression of therapeutically relevant photoreceptor and choroid trophic factors. (**a**) RT-qPCR analysis of mRNA levels of key RPE genes in mature E-RPE cells (>42 days in culture; *n* = 5). The results were normalized to an endogenous reference gene (*PPIA*) and are presented as mean fold change (2^−^^ΔΔCT^) relative to immature E-RPE culture (dotted line) ± standard deviation. Data were compared with a Mann–Whitney U test of ΔCt values; * *p* < 0.05; ** *p* < 0.01; NS: not significant. (**b**) Protein secreted by the post-confluent (PC) immature and mature E-RPE cultures into the conditioned media presented as mean concentration ± standard deviation (n = 5), Mann–Whitney U test.

**Figure 5 ijms-21-06066-f005:**
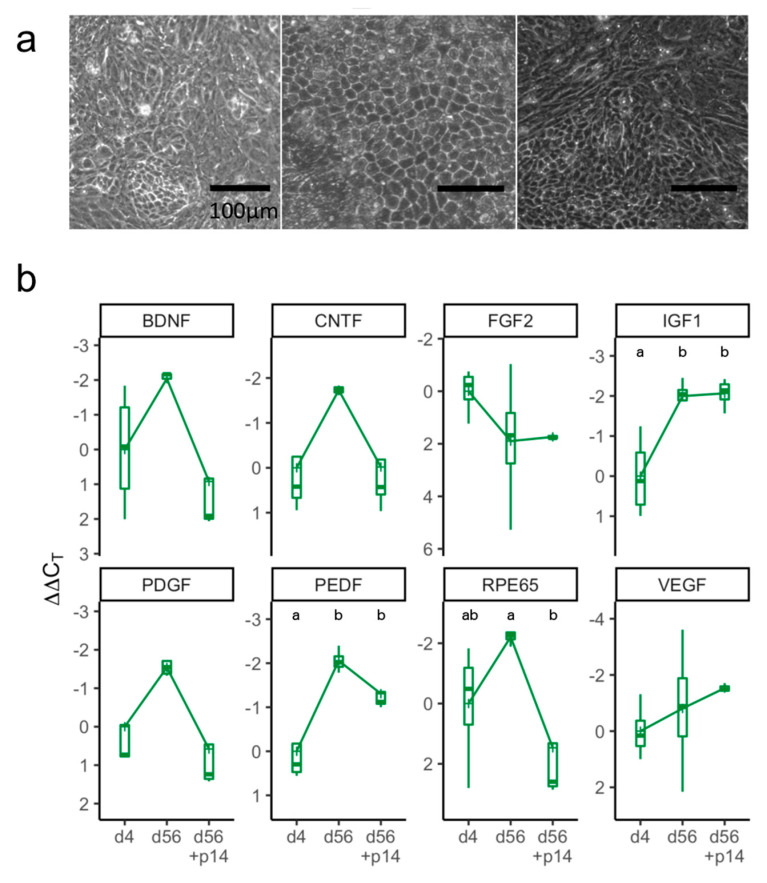
Markers of maturation are lost after passaging E-RPE cells. (**a**) Morphology of E-RPE cells at confluence, four days after seeding (left), at 56 days of maturation (center) and at a further 14 days after passaging (right). Scale bar represents 100 μm. (**b**) RT-qPCR analysis at three time points (4 days, 56 days and 56 + 14 days after passaging). Results were normalized to *PPIA* and represent mean ΔΔCt values (*n* = 3 or 4) ± standard deviation plotted on a negative *y*-axis (higher expression at the top). Kruskal-Wallis one-way ANOVA test was used to compare ΔΔCt values of the various maturation points within each gene followed by a Tukey’s honest significance test; different letters, *p* < 0.05.

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
