# Peer review of "In Vitro Maturation of Retinal Pigment Epithelium Is Essential for Maintaining High Expression of Key Functional Genes"

_ijms, 2020, doi:10.3390/ijms21176066_

Round 1

Reviewer 1 Report

Abdullah Al-Ani et al. realized a very well-written article consisting of several experiments evaluating that “In vitro maturation of retinal pigment epithelium is essential for maintaining high expression of key functional genes”. I consider the manuscript very fascinating but, in the same time, I suggest several revisions needed to improve the quality and the readability of the paper:

  • In 4.2 sub-section it could be described the statistical method applied to qRT-PCR analysis.
  • In 4.3 sub-section, the statistical analysis lacks of multiple comparison correction. I suggest the authors to perform FDR, Bonferroni or Tukey post-hoc test.
  • How did the authors choose specific genes to be evaluated in mature and immature RPE cells?
  • In “Discussion” section, as future perspective, I suggest the authors to add the possibility to realize an RNA-Seq experiment, in order to enforce results obtained in this work, especially focusing on further biomarkers related to RPE cell maturation and survival. Various papers already published could be used as reference, and three of the most updated and completed, that I suggest to cite, are PMID: 32413970
  • Finally, manuscript requires several English revisions and typos correction.

Author Response

Thank you very much for your insightful comments. Below we outline point-by-point how we have addressed and integrated your suggestions into the manuscript:

In 4.2 sub-section it could be described the statistical method applied to qRT-PCR analysis.

The details of the statistical methods are described in the figure legends, and generalized in section 4.5. We have added some more details to section 4.2 (lines 304-06).

In 4.3 sub-section, the statistical analysis lacks of multiple comparison correction. I suggest the authors to perform FDR, Bonferroni or Tukey post-hoc test.

We have adjusted our statistical analysis to correct for multiple comparisons by using Tukey post-hoc test. This adjusted statistical analysis is detailed in the Figure 5 legend (line 174).

How did the authors choose specific genes to be evaluated in mature and immature RPE cells?

We have carefully selected these 19 genes as they been extensively described in literature to influence RPE behaviour and function. These genes have been shown to influence RPE’s ability to support retinal photoreceptors, stabilize the choroid and perform as well as other key RPE functions. We had also included a brief explanation of our rationale for selecting these 19 genes (lines 181-86).

In “Discussion” section, as future perspective, I suggest the authors to add the possibility to realize an RNA-Seq experiment, in order to enforce results obtained in this work, especially focusing on further biomarkers related to RPE cell maturation and survival. Various papers already published could be used as reference, and three of the most updated and completed, that I suggest to cite, are PMID: 32413970

Thank you for this insightful point, and we agree with you that an RNA-Seq would further enforce our observations and perhaps define other differentially expressed genes. As such, we have added to our discussion the benefits of an RNA-Seq experiment to build on this work (lines 187-93).

Finally, manuscript requires several English revisions and typos correction.

After implementing the outlined revisions, multiple authors have thoroughly reviewed the manuscript and corrected several spelling and grammatical errors.

Thank you once again for your constructive feedback!

Reviewer 2 Report

In this paper the authors want to investigate specific changes that happen during RPE maturation and look at specific genes that should be upregulated during maturation. They compare ARPE-19 (a human cell line) and RPE derived from hESC.

The paper is well written, however I have a major issue concerning this paper as (i) I do not agree on the definition for maturation the authors are giving and (ii) authors should take a proper control, i.e. adult RPE.

(i) authors are defining maturation as a "cellular progress towards a fully-functional state as RPE cells integrate into a coherent tissue, as distinct from the properties exhibited by individual RPE cells that have been freshly juxtaposed". This definition is related to the formation of an epithelium and authors should investigate not only genes related to RPE but also proteins associated with junctions (ZO-1, Claudin,...) involved in the formation of this epithelium, and they shoul measure transepithelial resistance (that has been shown to be a marker for functional RPE). Moreover, they should look at Epithelial to Mesenchumal Transition (EMT) mecanism (specifically when passing RPE cells).

(ii) authors should take another control than ARPE-19 cells as, like they say, these cell do not exhibit all the markers of primary RPE cells. They should compare to adult RPE cells coming from eye tissue bank.

Other comments: in their introduction, authors should refere to two articles: Diniz et al., 2013 and Ben M'Barek et al., 2017 who directly compare transplantation of RPE as a tissu to dissociated cells and showing benefit in survival and function.

Author Response

Thank you very much for your insightful comments regarding our manuscript. Please find our point-by-point response to your comments below:

(i) authors are defining maturation as a "cellular progress towards a fully-functional state as RPE cells integrate into a coherent tissue, as distinct from the properties exhibited by individual RPE cells that have been freshly juxtaposed". This definition is related to the formation of an epithelium and authors should investigate not only genes related to RPE but also proteins associated with junctions (ZO-1, Claudin,...) involved in the formation of this epithelium, and they should measure transepithelial resistance (that has been shown to be a marker for functional RPE). Moreover, they should look at Epithelial to Mesenchymal Transition (EMT) mechanism (specifically when passing RPE cells).

We apologize for the lack of clarity in defining our aged cultures and the confusing wording of our conclusions. It was not our intention to claim that our RPE cultures are fully functional and coherent RPE tissues. We agree that the data we present in this manuscript are not sufficient to support this claim and investigations of tight junctions, TER and EMT mechanism would be required. While it may not have been clearly communicated, the intended central message of this manuscript is that we observe differential expression of key genes in RPE cells as a function of maturation. We hypothesize this differential gene expression pattern of RPE cells is due their presence in a supportive microenvironment (niche) or as a component of a monolayer. These observations are novel and can explain in part why RPE monolayers were matured in culture before being utilized for animal and human trials. To address your comments, we have edited the manuscript to remove references to tissue-level maturation, which we hope improves the clarity of our message (lines 56-8).

Moreover, we are also interested in learning if our matured RPE cultures possess some the key features of mature RPE tissue such as gap junction formation and establishing transepithelial resistance (TER). As such, we have incorporated your excellent experimental suggestions to the discussion section as future directions (lines 266-9). Unfortunately, we are unable to perform these experiments at this time given our limited laboratory access due to the coronavirus pandemic.

(ii) authors should take another control than ARPE-19 cells as, like they say, these cell do not exhibit all the markers of primary RPE cells. They should compare to adult RPE cells coming from eye tissue bank.

Thank you for your thoughtful suggestion. Our rationale for using ARPE-19 and E-RPE was to demonstrate that mature RPE differentially expresses key RPE genes across two distinct sources of RPE cells. While we agree that using primary RPE cells from the eye tissue bank would provide us with a control, against which we could compare gene expression and morphology, we are unable to acquire these tissues and perform the suggested experiment at this time. We acknowledge this limitation and have referenced studies showing broad transcriptional similarities between the cell lines we used and primary RPE (lines 248-51).

Other comments: in their introduction, authors should refere to two articles: Diniz et al., 2013 and Ben M'Barek et al., 2017 who directly compare transplantation of RPE as a tissu to dissociated cells and showing benefit in survival and function.

Thank you for directing our attention to these two interesting and relevant articles. We have added these references to the introduction and expanded our discussion of the benefits to transplanting a monolayer/sheet vs. single cell suspension (lines 60-1 and 264-6).

Round 2

Reviewer 1 Report

The authors revised their manuscript as suggested.

Reviewer 2 Report

Authors have made required corrections in their manuscript and their paper is now suitable for publication.